# Mechanisms of the Morphological Plasticity Induced by Phytohormones and the Environment in Plants

**DOI:** 10.3390/ijms22020765

**Published:** 2021-01-14

**Authors:** Gaojie Li, Shiqi Hu, Xuyao Zhao, Sunjeet Kumar, Yixian Li, Jingjing Yang, Hongwei Hou

**Affiliations:** 1The State Key Laboratory of Freshwater Ecology and Biotechnology, The Key Laboratory of Aquatic Biodiversity and Conservation of Chinese Academy of Sciences, Institute of Hydrobiology, Chinese Academy of Sciences, Wuhan 430072, China; ligaojie@ihb.ac.cn (G.L.); hushiqi@ihb.ac.cn (S.H.); zhaoxuyao@ihb.ac.cn (X.Z.); sunjeet@ihb.ac.cn (S.K.); liyixian@ihb.ac.cn (Y.L.); 2College of Modern Agricultural Sciences, University of Chinese Academy of Sciences, Beijing 100049, China

**Keywords:** environment, leaf, morphological plasticity, phytohormones, molecular mechanism

## Abstract

Plants adapt to environmental changes by regulating their development and growth. As an important interface between plants and their environment, leaf morphogenesis varies between species, populations, or even shows plasticity within individuals. Leaf growth is dependent on many environmental factors, such as light, temperature, and submergence. Phytohormones play key functions in leaf development and can act as molecular regulatory elements in response to environmental signals. In this review, we discuss the current knowledge on the effects of different environmental factors and phytohormone pathways on morphological plasticity and intend to summarize the advances in leaf development. In addition, we detail the molecular mechanisms of heterophylly, the representative of leaf plasticity, providing novel insights into phytohormones and the environmental adaptation in plants.

## 1. Introduction

Leaves are key interfaces between plants and their surrounding environment, functioning to capture sunlight, synthesize photosynthate, exchange gasses, sense ambient changes, and regulate their growth under heterogeneous conditions [1,2,3]. In part because of their sessile lifestyle, plants possess efficient systems of morphological plasticity and acclimation to environmental changes. The diversity of leaf shape, vein pattern, stomata, and other parameters not only vary among plants that belong to different species (Figure 1A) but also within a single plant [4,5,6] (Figure 1B). It is well known that the same genotype is capable of developing different phenotypes, which is regarded as the coordination of phenotype, development, and environment [7,8]. For example, heteroblasty was described as the changes in leaf shape during growth development [9], while anisophylly is coupled with asymmetry and leaf phyllotaxis [10]. Some species have even evolved the ability to develop significantly different leaf types under heterogeneous conditions, a phenomenon called heterophylly [11,12,13]. Furthermore, heteroblasty indicates the juvenile-to-adult transition marked by morphological changes, and it emphasizes the developmental stage-related plasticity [14]. However, heterophylly is an extreme morphological plasticity, which is induced by environmental conditions [12,13]. This morphological plasticity provides good models for studying leaf development. However, the mechanisms related to how plants sense environmental changes and develop final leaf forms is still not elucidated.

Given the rapid developments of plant functional genomics, many genes controlling leaf development have been studied, and the regulatory networks underlying these morphological processes have been well characterized [15]. Despite the fact that leaf development is related to genotype, the final shape is adjusted by environmental conditions, such as light, temperature, atmospheric carbon dioxide (CO_2_) concentrations, and submergence, to adapt to environmental variables [1,16]. The modulation of phytohormone signaling and distributions is a very effective strategy for quick environmental responses. Phytohormones are long-range molecular signals and have key functions in regulating plant growth and leaf development [11,17,18,19,20,21,22,23,24,25,26]. Thus, environmentally induced changes in hormone concentration, distribution, and/or sensitivity can promote coordinated developmental responses [27,28,29,30,31].

Here, we detailed the current knowledge on the molecular mechanisms underlying morphological plasticity regarding the environment, including environmental sensing, phytohormone signals, and leaf development in plants. Learning how plants use adaptive strategies in nature will help us to gain novel insights into plant science and further improve crops associated with a changing climate.

## 2. Environmental Sensing and Adaptation to Light and Temperature

Photosynthesis efficiency depends on the light capture of leaves. As a result, the balance of maximizing light capture and minimizing the harmful impact of high light is a coordinated developmental response. For example, plants prefer to develop broad leaves to maximize light capture, but if the sunlight is too harsh it may lead to overheating and cause harm to the plants [32,33]. In contrast, leaf development also responds to shade (a reduction in the red (660 nm) to far-red (730 nm), R/FR), which is called shade avoidance syndrome (SAS), showing petioles elongation, leaf upward bending, and leaf area decreasing [34,35] (Figure 2A). The upward movement of the leaves allows the plant to elevate the position of the foliage in order to maximize light capture [34,36]. Other aspects are also affected by light, such as leaf complexity, stomata density, and leaf thickness, which increased in the high light conditions [16,37,38,39]. In *Rorippa aquatica* (Brassicaceae), leaf complexity is dramatically increased in high light conditions [40]. In some other species, such as *Nuphar lutea* (Nymphaeaceae), *Rumex palustris* (Polygonaceae), and *Hygrophila difformis* (Acanthaceae), light change even induced the rearrangement of chloroplasts and altered the photosynthetic biochemistry to adapt the plant to aquatic conditions [41,42,43]. The photoperiod also significantly regulates leaf form. For example, short daylength induced submerged leaves, while long daylength induced terrestrial leaves of *P. palustris* and *Ranunculus aquatilis* (Ranunculaceae) [44,45].

Increasing surrounding temperature affects numerous developmental traits among plants, and the morphological changes that occur in plants in response to temperature changes are called thermomorphogenesis [46,47,48]. In order to adapt to high temperatures, plants developed elongated hypocotyls and petioles, as well as a decrease in leaf thickness and an increasing stomatal density [47,49,50]. These morphological responses are believed to cool plants and reduce the damage caused by sunlight through the upward bending of leaves [46,51,52]. Leaf dissection has for a long-time been thought to correlate with ambient temperature [5]. For example, plants growing in cold climates tend to develop serrated or deep-lobed leaves, while plants growing in warm conditions display shallow-lobed leaves [53,54,55,56,57]. To some degree, leaf dissection was used as an indicator for predicting paleoclimate [5,58]. The change in temperature of a single leaf of *R. aquatica* affects the epidermal cell size in developing leaves, and hence the morphology of the whole plant is affected [1,59]. In *Ludwigia arcuata* (Onagraceae), low temperature induced the elongation of epidermal cells and thus lead to the aquatic leaf form [60]. It was recently verified that pectin and cortical microtubules drive morphogenesis in plant epidermal cells [61,62], but how these epidermal changes are regulated by temperature is still unknown.

It was verified that auxin signal functions to connect temperature sensing with growth responses in hypocotyls [63,64]. In *Arabidopsis thaliana*, temperature changes can be sensed by the inactivation of photoreceptors such as phytochrome B (phyB), whose function in thermoregulation operates via the PHYTOCHROME-INTERACTING FACTOR 4 (PIF4) for high temperature-induced hypocotyl elongation [65,66]. High temperature-activated PIF4 directly upregulates the expression of auxin biosynthesis genes (e.g., *YUCCA8*, *TAA1*, and *CYB79B2*), and as a result, the accumulated auxin induces hypocotyl elongation and leaf hyponasty [67,68]. High temperature also induced PIF4 expression by inactivating EARLY FLOWERING 3 (ELF3) that directly represses PIF4. In high temperature conditions, ELF3 binding to the *PIF4* promoter is decreased, and thus PIF4 was activated for thermomorphogenesis [69,70]. Auxin could theoretically induce elongation growth; however, it was recently reported that the phytohormone brassinosteroid (BR) activates elongation growth downstream of auxin to act in themomorphogenesis [71,72].

Temperature and light signals are integrated into the PIF and the relevant genetic network, which controls auxin biosynthesis [67,73]. Photomorphogenesis and shade avoidance responses, including stem/hypocotyl elongation are mediated by PIF4 [74]. The stability of the PIF4 protein is regulated by light, and it is dephosphorylated and stable in the dark, while it is rapidly phosphorylated by phyB-mediated signaling and degradation upon red light irradiation [74]. Interestingly, although phyB and PIF4 antagonistically regulate photomorphogenesis and shade avoidance responses, they cooperatively promote stomatal development in response to high light [39]. Shade also induces the expression of gibberellic acid (GA) biosynthetic enzymes and leads to an accumulation of GA, which then promotes the degradation of DELLAs. It was found that DELLA directly interacts with PIF4 and prevents it from binding to target promoters [75,76]. Besides, the ethylene response also shows short hypocotyls, short roots, and an exaggerated apical hook [77]. PIF4 also promotes ethylene biosynthesis by activating the expression of ethylene biosynthesis genes (e.g., *ACS2*, *6*, *8*, and *9*) and enhances ethylene signaling by activating the transcription factor ETHYLENE INSENSITIVE 3 (EIN3) [78,79].

Light and temperature are the most critical environmental factors for plant growth, and even a slight change can lead to disasters of plants [80,81]. We mentioned above that PIF4 may be a key element that functions in the light/temperature-dependent morphological plasticity and the crosstalk of phytohormones such as auxin, ethylene, and GA. Future studies based on these gene pathways and phytohormones will not only reveal novel mechanisms on the light and temperature response but will also have implications on crop improvement through use of these plastic strategies.

## 3. Environmental Sensing and Adaptation to Submergence

Under flooding or submerged conditions, plants find it difficult to obtain enough O_2_ for respiration. Terrestrial plants, such as *A. thaliana* and *Solanum lycopersicum* (tomato), which are intolerant to flooding, find that submerged conditions induce their leaves to turn pale and suppresses their plant growth [73,82]. Deepwater rice survive periodic flooding and consequent oxygen deficiency by activating an internode growth of stems to keep above the water [83] (Figure 2B). In other species such as *R. palustris*, elongated leaves and decreased thickness helps the plant to obtain a relatively increased gas exchange under submerged conditions [41]. In some aquatic, dimorphic types of plants, their submerged leaves are always thin, narrow, or dissected and contain fewer stomata, while aerial leaves are thick, broad, and entire, and have more stomata [12,40,84,85]. Although narrow or dissected leaves are less efficient at absorbing sunlight than those with wider blades, they can better withstand the destructive force of water flow and more efficiently incorporate CO_2_ and mineral nutrients than entire leaves [86,87,88].

ABA and ethylene are key regulators of drought and submerge response, separately. ABA was regarded as a stress hormone, which accumulates rapidly in response to drought/dehydration stress and plays a crucial role in stomatal closure, root growth, and the production of protective metabolites [20,89]. ABA levels in unstressed plants are low, but accumulated highly under reduced water potentials by the activation of key synthesis genes *9-cis-epoxycarotenoid dioxygenases* (*NCEDs*) [90]. Upon perception of ABA, the ABA receptor pyrabactin resistance 1 (PYR1)-like protein PYL, regulatory components of the ABA receptor (RCAR) proteins, inhibit the activity of clade A protein phosphatase type 2C (PP2C) phosphatases, thus releasing the subclass III sucrose nonfermenting 1-related kinase 2 (SnRK2s, including SnRK2.2, SnRK2.3, and SnRK2.6) to phosphorylate downstream proteins [91,92]. The arabidopsis protein kinases SnRK2s function as central and positive regulators of the ABA signaling pathway and are involved in stomatal closure, osmotic stress responses, and have an evolutionarily conserved function on plant adaptation to the terrestrial environment [93,94,95].

Aquatic plants, such as rice, have evolved adaptive mechanisms to survive under submergence. When subjected to flooding, rice or deepwater rice accumulates high ethylene, which activates gibberellin biosynthesis gene *SEMIDWARF 1* (*SD1*), promotes GA-dependent elongation, and results in an “escape” strategy to reestablish contact with the air [83]. Recent studies have found that the submergence-induced GA accumulation activates *ACCELERATOR OF INTERNODE ELONGATION 1* (*ACE1*), which confers cells of the intercalary meristematic region with the competence for cell division, leading to internode elongation in the presence of GA. In contrast, high GA repressed *DECELERATOR OF INTERNODE ELONGATION* 1 (*DEC1*) suppresses internode elongation, whereas downregulation of DEC1 allows internode elongation [96]. Under submerged conditions, ethylene also induces the expression of two ethylene response factors (ERFs), *SNORKEL1* (*SK1*) and *SK2*, to trigger remarkable internode elongation via GA [97]. However, the response may vary between species, as GA levels in *Rumex acetosa* remain unchanged, although ethylene increased during submergence [98]. For the submergence of terrestrial plants, such as *A. thaliana*, the limited gas diffusion causes passive ethylene accumulation, leading to ETHYLENE INSENSITIVE 2 (EIN2) and EIN3/EIN3-like 1 (EIL1)-dependent signaling and enhanced production of the nitric oxide (NO) scavenger PHYTOGLOBIN 1 (PGB1). The enhanced PGB1 levels lead to NO depletion, enhancing group VII ethylene response factor (ERFVII) stability [99]. The constitutively synthesized ERVIIs (e.g., RELATED TO APETALA 2.12 (RAP2.12), RAP2.2, and RAP2.3) redundantly act as the principal activators of many hypoxia adaptive genes and lead to flooding survival [43].

Phytohormone signals also play key roles in leaf development. For example, the recruitment of leaf founder cells in the shoot apical meristem (SAM) is mediated by the formation of a concentration maxima of auxin [100,101]. Altering the endogenous auxin levels and localization results in leaf simplification in a tomato plant, while downregulating auxin biosynthesis genes (e.g., *YUCCA*) was reported to inhibit organ initiation in many species such as *Arabidopsis*, maize, and petunia [102,103,104]. Cytokinin (CK) also plays an important role in SAM maintenance [105,106,107]. Overexpression of the CK biosynthesis genes in tomato leaves leads to the formation of highly compound leaves. However, exogenous application of CK causes minor leaf phenotypes in the tomato [108]. Increasing GA levels in tomatoes result in tall plants with larger and simpler leaves [109]. Interestingly, this GA response is not common, and in some species, GA has the opposite effect of inducing more compound leaves [110,111]. To better understand the relationship of phytohormones and leaf development, in the next section we will discuss the molecular mechanisms of leaf development.

## 4. Mechanisms of Leaf Development: The Gene Regulatory Networks (GRNs)

Despite the diversity of leaf shapes, the molecular mechanisms of leaf development in most species are shared [50,107,112,113]. Recently, the complexity of the genetic networks controlling leaf development was fully summarized [2,114]. Here, we briefly review a classic view of the regulatory pathway which operates in leaf development, in order to better understand the mechanism of leaf plasticity.

Leaves are initiated at the flank of the SAM, which contains a pool of undifferentiated cells at the plant aerial apex [114,115] (Figure 3A). PIN-FORMED 1 (PIN1), the auxin efflux carrier, dynamically repolarizes and creates directional auxin flows at specific positions in the SAM. Auxin locally repressed the expression of *class-I KNOTTED-LIKE HOMEOBOX* (*KNOXI*) genes, which are responsible for stem cell maintenance in the SAM, like *SHOOTMERISTEMLESS* (*STM*) and *BREVIPEDICELLUS* (*BP*) [101,115]. *ARP* genes (including *ASYMMETRIC LEAVES 1* (*AS1*), *ROUGHSHEATH 2,* and *PHANTASTICA*) like AS1 interact with ASYMMETRIC LEAVES 2 (AS2), and their protein complexes bind directly to the promoter of *KNOXI* genes and repress their expression [116,117,118]. The formation of an auxin gradient within the SAM also contributes to the formation of boundary domains that separate primordia from the rest of the meristem [114]. These domains are maintained by the activity of several transcription factors, such as the NO APICAL MERISTEM/CUP-SHAPED COTYLEDON (NAM/CUC) family [119]. KNOXI transcription factors maintain the meristematic activity in the SAM through CK and GA, by activating the CK biosynthesis gene *ISOPENTENYLTRANSFERASE 7* (*IPT7*), which maintains cell proliferation while preventing cell differentiation by repressing its biosynthesis gene *GA 20-oxidase* (*GA20ox*) and activating the deactivation gene *GA2ox* [22,120].

Starting on the flank of the SAM, the newly initiated leaf primordia becomes asymmetric in three axes: the adaxial-abaxial, medial-lateral, and proximal-distal [122] (Figure 3B). Among these, the adaxial-abaxial polarity allows the further establishment of lateral polarity [114]. To establish adaxial-abaxial polarity, *HD-ZIPIII* genes expressed in the adaxial side of leaf primordia, function antagonistically to *KANADI* (*KAN*) genes, which are expressed in the abaxial side. *YABBY* (*YAB*) functions relatively later in leaf development and acts downstream of *KAN* genes on the abaxial side [124,125,126,127] (Figure 3B). MicroRNAs, like miR165/166, are also expressed towards the abaxial side, negatively regulating *HD-ZIPIII* to restrict its expression to the adaxial side of leaf primordia [128,129,130]. In contrast, AS1 and AS2 promote the expression of *HD-ZIPIII* on the adaxial side and repress the expression of miR165/166, *KAN*, and *YAB* genes [131,132]. Trans-acting short interfering RNAs (ta-siRNAs), whose targets are miR165/166 and Auxin response factors (ARFs) such as ARF3 and ARF4 transcription factors, are generated in the adaxial side and restrict the expression of ARF3/4 genes to the abaxial side [133,134,135].

The proximal-distal axis may be genetically established when a leaf primordium emerges from the shoot apex (Figure 3C). During this process, *KNOXI* genes are expressed in the boundary region and *CUC* genes, which are negatively regulated by miR164, have positive feedback with *KNOXI* [136,137]. KNOXI and AS1 appear to be involved in the proximal-distal polarity patterning, and *BLADE ON PETIOLE* (*BOP*) genes, such as *BOP1* and *BOP2*, are expressed in the proximal region, directly repressing *KNOXI* or indirectly restricting the location of KNOXI by activating *AS2* [118,138,139]. In addition, ARF3/4 genes also repress KNOXI to promote organogenesis at the shoot apex [140]. The mechanism of medio-lateral polarity is still not clear, and only a few studies have found several regulatory genes. KANs and HD-ZIPIII antagonize each other and inversely regulate the genes involved in auxin transport and biosynthesis, resulting in a high auxin level in the abaxial domain [125,141,142]. The high abaxial auxin and the adaxial expression of *MONOPTEROS* (*MP*) results in higher auxin response, therefore, it induced the activation of the *WUSCHEL- related homeobox* (*WOX*) genes, *WOX1* and *PRESSED-FLOWER* (*PRS*) [141,142]. The expression of *WOX1* and *PRS* is restricted to the middle domain but highly expressed in the marginal region, promoting the formation of serration or leaflets [50,121,143,144,145].

After leaf blade initiation, leaves grow according to two main processes based on cell division and expansion (Figure 3D). Two classes of miRNA/transcription factors play antagonistic roles in cell proliferation and differentiation for subsequent leaf development (Figure 3D). The class II TEOSINTE BRANCHED1/CYCLOIDEA/PROLIFERATING CELL FACTOR (TCP), which are downregulated by miR319, function to promote cell differentiation and expansion in the distal part of leaves [146], while GROWTH REGULATING FACTORS (GRFs), which are repressed by miR396, function with GRF interacting factors (GIFs) to promote cell proliferation in the proximal ends of leaves [147,148,149]. Class II TCP, like TCP4, can also directly activate miR396 to inhibit the expression of *GRF* targets or repress the expression of *GRF/GIF* genes via unknown mechanisms [148,150,151]. *CUC* genes play key roles for marginal morphogenesis and are repressed by Class II TCP and miR164 [152,153,154]. In addition, PRS is also repressed by class II TCP and NGATHA (NGA), promoting cell proliferation in the leaf margin [155]. Recent studies also found that WOX1 regulates Class II TCP at both the transcriptional and translational level and regulates leaf size and vein pattern in *Cucumis sativus* [156].

There are significant differences in simple-leafed and compound-leafed species (Figure 3E). Although in some leguminous lineages *LEAFY* (*LFY*) activity modulates leaf complexity [157], *KNOXI* genes are the key factors regulating leaf morphological differences among species [40,115]. In simple-leafed species like *A. thaliana*, *KNOXI* is only expressed in the SAM, and marginal serrations are modified by the feedback regulation of auxin maxima and *CUC* genes [158]. Correspondingly, in compound-leafed species like tomato and *Cardamine hirsute*, *KNOXI* is re-activated in the leaf primordia, which results in the formation of leaflets by the feedback regulation of auxin maxima, *KNOXI*, and *CUC* genes [101,123,159].

## 5. Molecular Mechanisms of Heterophylly—A Representative of Leaf Plasticity

Environmental signals were integrated into GRNs and subsequently induced rapid and acclimated changes. Morphological plasticity was found in many species, as shown by changing leaf size, shape and thickness, and stomatal density, which has been seen in several species under different conditions or development stages [50,112,113,160]. Among those, heterophyllous plants show extreme plasticity in response to environmental factors, and were regarded as an ideal system for studying environmentally induced leaf plasticity [3,11,86]. To achieve various leaf patterns with different environments, heterophyllous plants have evolved diverse mechanisms for leaf development (Figure 4). The first illustrated example is *R. aquatica*, which develops deeply dissected leaves under submerged or low temperature conditions, while it has shallow serrated leaves under terrestrial or high temperature conditions [10]. It was found that the expression levels of *KNOXI*, which is the key gene that decides the final leaf form in many plant species, upregulated in submergence and low temperature but downregulated in terrestrial and high temperature conditions. Thus, due to the conserved function of *KNOXI* in activating CK and repressing GA accumulation (see Section 3 and Section 4), the phytohormone pattern changes in leaf primordia therefore regulate the final leaf shape [40]. Interestingly, *R. aquatica* also develops deeply dissected leaves in high illumination and shows high expression of *KNOXI*, indicating a potential relationship of *KNOXI* and light response [40].

In another heterophyllous plant *Ranunculus trichophyllus*, ABA and ethylene mainly control terrestrial and aquatic leaf development, respectively. In terrestrial conditions, high ABA induced the ABSCISIC ACID INSENSITIVE 3 (ABI3)-mediated activation of adaxial genes (e.g., *HD-ZIPIII*), which then increased the expression of STOMAGEN (*STO*) and *VASCULAR-RELATED NAC-DOMAIN 7* (*VND7*), resulting in increased stomata density and vessel elements. In contrast, submerged conditions activated ethylene synthesis and accumulation, which then induced the expression of EIN3-mediated activation of abaxial genes (e.g., *KAN*) and repressed *STO* and *VND7*, resulting in a lack of stomata and reduced vessel development in submerged leaves [12]. Recently, studies on *Potamogeton wrightii* (heterophyllous) and its sister species *P. perfoliatus* (homophyllous) have shown that exogenous ABA application induced stomata in both submerged species, *P. perfoliatus* as well as in heterophyllous *P. wrightii* [85]. However, under salinity stress, which promotes ABA biosynthesis by NCEDs, stomata were only induced in *P. wrightii*, but not in *P. perfoliatus*. These results suggested that differences in the ABA-mediated stress responses were responsible for the variation in morphological plasticity between the two *Potamogeton* plants under natural conditions [85].

Morphological plasticity in the genus *Capsella*, such as the increased leaf complexity induced by low temperatures, is mediated by the activation of *REDUCED COMPLEXITY* (*RCO*) [161]. Recent studies in *A. thaliana* and its relative *C. hirsuta* have shown that the different leaflet development also requires *RCO*, evolved in the Brassicaceae family through gene duplication, and was lost in *A. thaliana*, contributing to leaf simplification in this species [161,162]. *RCO* functions specifically in leaf development, where it represses the cell growth at flanks [161,162]. Subsequently, researchers have found that differences in the leaf originate from two distinct processes that act in the *C. hirsuta*, but not in the *A. thaliana* leaves. Firstly, *KNOXI* gene (e.g., *STM*) delayed differentiation but increased the size and number of leaf protrusions. Secondly, *RCO* leads to growth differences created by the inhibition of marginal patterning [163]. *RCO* also coordinates the homeostasis of the phytohormone CK through CK biosynthesis and catabolism and their coordinates are essential for complex leaf development in *C. hirsuta* [164,165]. However, whether the morphological plasticity in *C. grandiflora* is achieved by the *RCO*/CK module is still unknown.

Even though heterophylly has been seen for centuries, the molecular mechanisms of these plants are still largely unknown. Recent advances in omics technologies and gene transformation have allowed genetic analyses of many heterophyllous species, which make it possible to investigate the mechanisms of plant development, morphological plasticity, and environmental adaption [3,13]. For example, *Potamogeton octandrus* is an aquatic heterophyllous plant that has ovate and flat floating leaves, but narrow and thin submerged leaves. Transcriptome analyses have found that many of the different expression genes (DEGs) were found in the “plant hormone signal transduction” category and endogenous levels of hormones such as ABA, cytokinin, GA, and auxin changed between conditions [16]. Comparative transcriptomics also reveals genes related to physiological adaptions of two accessions of *R. aquatica*, indicating that different genotypes might develop a novel strategy for adaptation [166].

Based on the above, we have found that key genes and phytohormones function in leaf development, and environmental responses play an important role in leaf plasticity [167,168,169]. There are still questions: If key genes involved in heterophylly (e.g., *KNOXI*) have conserved roles among species, why are some plants able to develop heterophylly for environmental adaption while others cannot? Do non-coding RNA and cis-acting regulatory elements function to regulate the morphological plasticity?

## 6. Future Perspectives

In a rapidly changing climate, plants are facing great challenges from the environment. Recent advances in omics technologies and gene transformation have allowed genetic analyses to investigate the molecular mechanisms of plant development, ecology, and evolution [3,170]. Recent works also have made great breakthroughs in the fields of environmental signals sensing [171,172,173], phytohormone interactive networks [29], and plant stress combinations [174,175]. Until now, the molecular mechanisms underlying leaf development have been extensively elucidated [114,176]. However, the large number of mechanisms of environmentally induced leaf plasticity are still unknown, which limits the application of morphological variety in plant improvement. Thus, the identification of key genes from genomics, transcriptomics, and phenomics, or CRISPR-mediated gene editing, is also a powerful and efficient approach to discover the novel mechanisms underlying plant environmental adaptation. Identifying the developmental and genetic basis of leaf plasticity induced by environmental changes will be important to engineer more adaptive crops in the face of future global change.

## Figures and Tables

**Figure 1 ijms-22-00765-f001:**
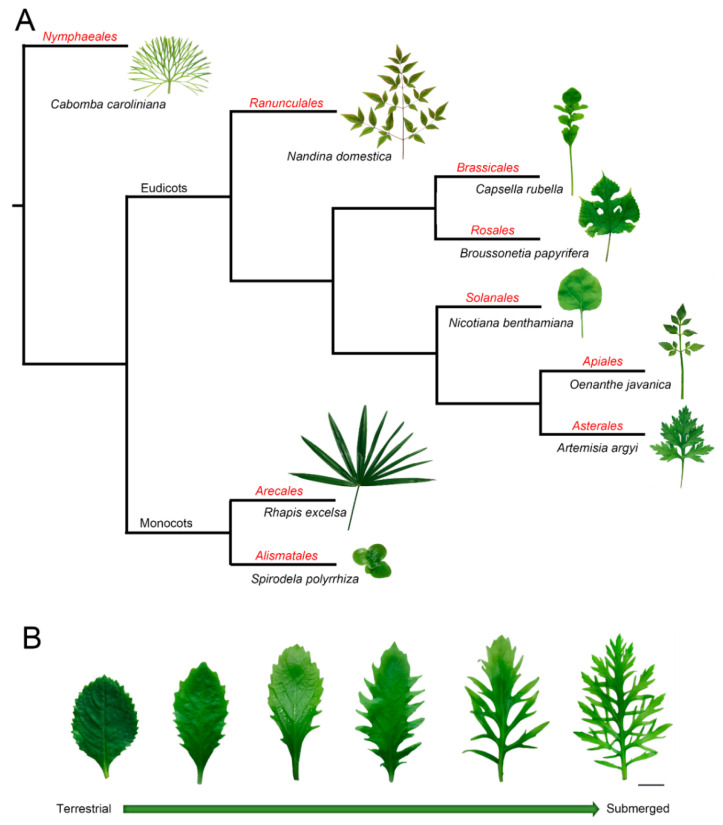
The phylogeny and typical leaf shape among plant species. (**A**) The phylogeny and typical leaf shape among species from different orders. Red text indicates the order name. (**B**) Leaves from a heterophyllous plant (*Hygrophila difformis*) shifted from terrestrial to submerged conditions. Successive leaves are in phyllotactic order. Bar = 1 cm. All photos were taken by the camera (Canon EOS80D, Japan) and plant materials were collected from the Key Laboratory of Aquatic Biodiversity and Conservation of Chinese Academy of Sciences (Institute of Hydrobiology, Chinese Academy of Sciences). The phylogenetic tree was based on the online software “Phylomatic” (http://phylodiversity.net/phylomatic/).

**Figure 2 ijms-22-00765-f002:**
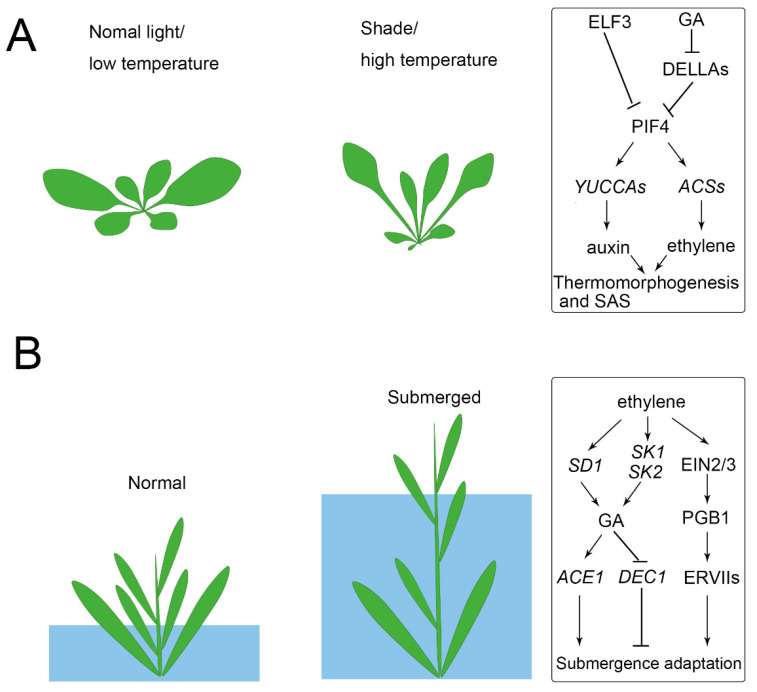
Example of plant developmental responses to environmental changes. (**A**) Both shaded light and an increase in temperature induce the elongation of the petiole, a reduction of leaf area, and an upward movement of the leaves. ELF3 directly represses PIF4, and this repression was released in shade/high temperature conditions. PIF4 activates auxin synthesis by upregulating *YUCCAs* and activating ethylene synthesis by upregulating *ACSs* for thermomorphogenesis and shade avoidance syndrome (SAS). Shade/high temperature also induces high levels of gibberellic acid (GA) and the degradation of DELLAs, which therefore release PIF4 for binding to target promoters. (**B**) Deepwater rice activates stem elongation growth depending on the water level. Submerged conditions accumulate high ethylene and activate *SD1* for GA synthesis. GA promotes stem elongation through the activation of *ACE1* and repression of *DEC1*. Ethylene also induced EIN2/EIN3 signaling and thus enhanced PGB1 to improve ERFVII stability for flooding survival.

**Figure 3 ijms-22-00765-f003:**
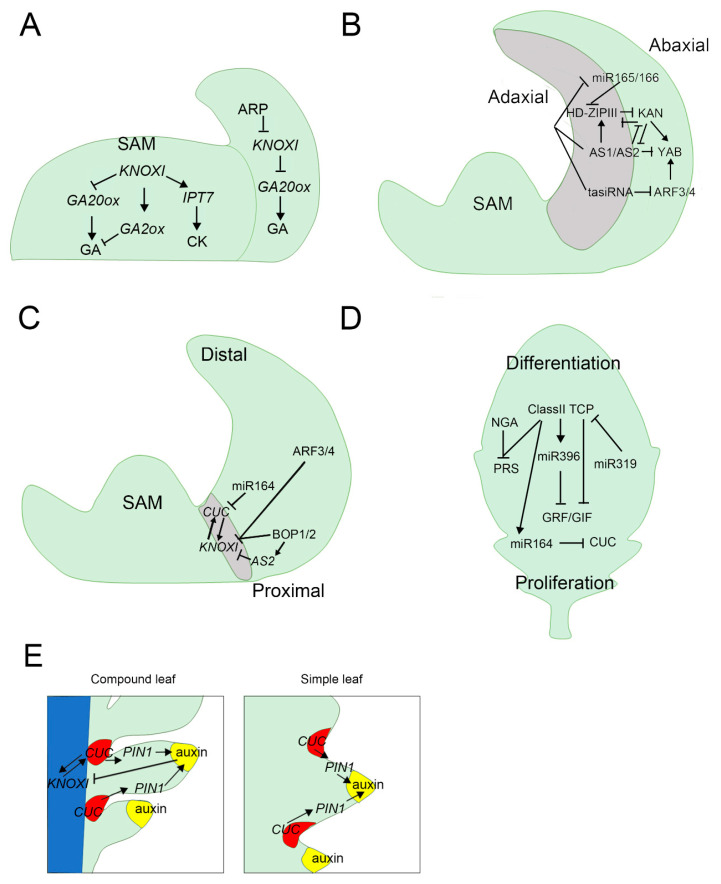
Genetic and hormonal factors that control leaf development. (**A**) Genetic and hormonal factors are controlling primordium development. Class-I KNOTTED-LIKE HOMEOBOX (KNOXI) proteins maintain high cytokinin (CK) levels and low GA levels in the shoot apical meristem (SAM). ARP maintains high GA level through repression of KNOXI. (**B**) Adaxial-abaxial polarity establishment in a developing leaf. HD-ZIPIII functions antagonistically to KANADI (KAN) and *YABBY* (*YAB*) acts downstream of KAN on the abaxial side. miR165/166 represses HD-ZIPIII, but ASYMMETRIC LEAVES 1 (AS1) and AS2 promote the expression of *HD-ZIPIII* on the adaxial side and repress miR165/166, *KAN*, and *YAB*. ta-siRNAs target miR165/166 and Auxin response factor 3/4 (ARF3/4) to restrict them to the abaxial side. (**C**) Proximal-distal polarity establishment in a developing leaf. KNOXI genes are expressed in the boundary region, and CUP-SHAPED COTYLEDONS (CUCs) have positive feedbacks with KNOXI. Blade on PETIOLE 1 (BOP1) and BOP2 are expressed in the proximal region to repress KNOXI directly, or indirectly by AS2. ARF3/4 also repress KNOXI to promote organogenesis at the shoot apex. (**D**) The switch from cell proliferation to differentiation follows a process that is promoted by the miR319-TCP module and repressed by the miR396-GRF module. *PRESSED FLOWER* (*PRS*) is also repressed by class II TCP and NGATHA (NGA), promoting cell proliferation in the leaf margin. (**E**) Common molecular pathways underpin both simple and compound leaf formation. PIN-FORMED 1 (PIN1) localization at the developing leaf is polar so that an auxin activity maximum is formed at the tip of both serration and leaflet. *KNOXI* are expressed in the rachis of the compound leaf, where they activate *CUC* expression at the distal boundary of the leaflet and promote polar localization of PIN1 in the leaflets. In turn, *CUC* activity maintains *KNOXI* expression in the rachis while auxin downregulates *KNOXI* expression for leaflet formation. *CUC* expression and auxin maxima promote the development of serrations. Yellow represents an auxin activity maximum, red the domain of *CUC* expression, and the blue color denotes the expression domain of *KNOXI*. Panel A, C, and D is based on [121] and B is based on [122]. Panel E is based on [123].

**Figure 4 ijms-22-00765-f004:**
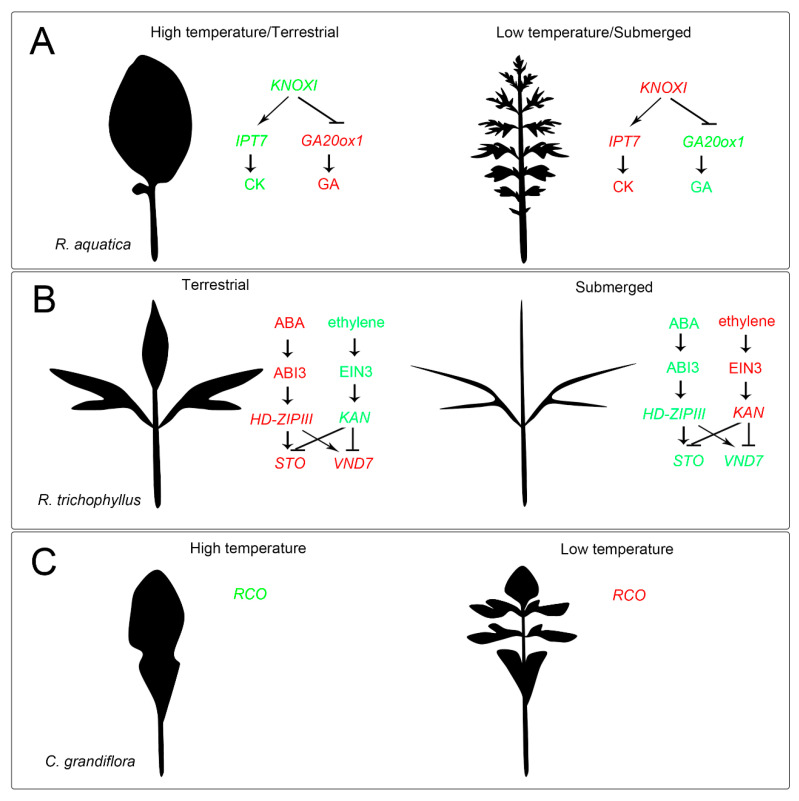
Molecular mechanisms of heterophylly. (**A**) The mechanism of heterophylly in *R. aquatica*. Complex leaves were induced by the upregulated *KNOXI* and thus induced repression of *Ga20ox1* and downregulated GA, while simple leaves were induced by the downregulated *KNOXI* and thus induced upregulated *Ga20ox1* and GA. *KNOXI* also induced the accumulation of CK by the regulation of *ISOPENTENYLTRANSFERASE 7* (*IPT7*). (**B**) The mechanism of heterophylly in *R. trichophyllus*. Terrestrial conditions induced ABA accumulation and activates HD-ZIPIII-mediated *STOMAGEN* (*STO*) and *VASCULAR-RELATED NAC-DOMAIN 7* (*VND7*) upregulation via ABI3, while submerged conditions induced ethylene accumulation and activate KAN-mediated *STO* and *VND7* downregulation via *EIN3*. (**C**) The heterophylly of *C. grandiflora* was induced by the temperature, dependent on *REDUCED COMPLEXITY* (*RCO*). Red in (**A**–**C**) represents upregulated genes or accumulated phytohormones, and green in (**A**–**C**) represents downregulated genes or phytohormones.

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
