# Peer review of "Mechanisms of the Morphological Plasticity Induced by Phytohormones and the Environment in Plants"

_ijms, 2021, doi:10.3390/ijms22020765_

Round 1

Reviewer 1 Report

Comments to the Author

Li et al. review what is known about heterophylly, a manifestation of phenotypic plasticity in plants resulting in leaves with different morphologies depending on the environment, and its hormonal regulation. It is a fascinating phenomenon, and it is certainly worthwhile to summarize what is known about its molecular underpinnings. The authors have done that quite exhaustively, focusing on different model systems of heterophylly and how different hormones and molecular mechanisms are involved. The text provides a clear roadmap to the topic and offers future directions of research. However, sometimes it looks like just a list of previous studies. I would like to suggest that the authors discuss the relationship between heterophylly and each topic or add a brief future perspective at the end of each section.

I have a few minor suggestions for the authors to consider.

Minor comments

The contrast with heteroblasty might be very useful. Please add some sentences to explain the difference between heterophylly and heteroblasty in the introduction.

What do the authors say in Figure 1A? It is not clear. Please revise the manuscript or remove the Figure 1A.

In Figure 1A, please label clade names in addition to scientific names. It makes the phylogeny more informative.

p3, L67; Please replace “high” with “harsh”.

p3, L78; Please replace “Rumex Palustris” with “Rumex palustris”.

p5, 141; Any conclusion? The authors extensively reviewed the relationship among light, temperature, PIF4, auxin, and ethylene. But it looks like just a list of previous studies. Please add a brief conclusion about the relationship between heterophylly and what the authors discussed above.

p6, L171; This also needs a brief conclusion.

p6, L184; Please cite “Mitsui Y, Nomura N, Isagi Y, Tobe H, Setoguchi H. Ecological barriers to gene flow between riparian and forest species of Ainsliaea (Asteraceae). Evolution. 2011 Feb;65(2):335-49. doi: 10.1111/j.1558-5646.2010.01129.x. Epub 2010 Oct 7. PMID: 20840597.”

p10, L330; The authors state “Phenotypic plasticity is shared by all plants, ~”. However, it seems to be difficult to assert that. Please revise the sentence.

p10, Figure 4A; Please replace “R. aqquatica” with “R. aquatica”.

p11, L374, 376, and 379; Please italicized “A. thaliana”.

p11, L399; Please remove an extra “.”

p11, L399; This also needs a brief conclusion of this section.

Reviewer 2 Report

All my comments are in the Review.
